# Ruthenium Complex HB324 Induces Apoptosis via Mitochondrial Pathway with an Upregulation of Harakiri and Overcomes Cisplatin Resistance in Neuroblastoma Cells In Vitro

**DOI:** 10.3390/ijms24020952

**Published:** 2023-01-04

**Authors:** Nicola L. Wilke, Hilke Burmeister, Corazon Frias, Ingo Ott, Aram Prokop

**Affiliations:** 1Department of Pediatric Oncology/Hematology, Helios Clinics Schwerin, Wismarsche Straße 393–397, 19049 Schwerin, Germany; 2Medical School Hamburg (MSH), University of Applied Sciences and Medical University, Am Kaiserkai 1, 20457 Hamburg, Germany; 3Department of Pediatric Hematology/Oncology, Children’s Hospital Cologne, Amsterdamer Straße 59, 50735 Cologne, Germany; 4Institute of Medicinal and Pharmaceutical Chemistry, Technische Universität Braunschweig, Beethovenstr. 55, 38106 Braunschweig, Germany

**Keywords:** cancer chemotherapy, ruthenium, leukemia, lymphoma, neuroblastoma, multidrug resistance, apoptosis, mitochondrial pathway, Harakiri

## Abstract

Ruthenium(II) complexes with N-heterocyclic carbene (NHC) ligands have recently attracted attention as novel chemotherapeutic agents. The complex HB324 was intensively studied as an apoptosis-inducing compound in resistant cell lines. HB324 induced apoptosis via mitochondrial pathways. Of particular interest is the upregulation of the Harakiri resistance protein, which inhibits the anti-apoptotic and death repressor proteins Bcl-2 (B-cell lymphoma 2) and BCL-xL (B-cell lymphoma-extra large). Moreover, HB324 showed synergistic activity with various established anticancer drugs and overcame resistance in several cell lines, such as neuroblastoma cells. In conclusion, HB324 showed promising potential as a novel anticancer agent in vitro, suggesting further investigations on this and other preclinical ruthenium drug candidates.

## 1. Introduction

The biggest problems in cancer therapy, especially in pediatric oncology, are the therapeutic side effects and the development of resistance to the conventional cytostatic drugs used [1,2,3]. Therefore, research has focused on substances that selectively induce apoptosis in malignant cells and at low concentrations. We and others have reported on novel metal-based compounds that can overcome these resistances and have a low toxicity [4,5,6,7,8,9,10,11]. Metal complexes have long been of big interest in modern cancer research [12,13]. The inorganic platinum complexes cisplatin and oxaliplatin have proven to be very effective and have become established in cancer therapy [14,15]. Due to tissue toxicity, side effects, and the development of resistance, there is a strong interest in the development of new effective complexes with other metals. Ruthenium compounds may be a possible alternative due to lower toxicity and different mechanisms of action [16,17,18]. Therefore, we investigated two different ruthenium complexes and their non-metallic (non-ruthenium) ligands (Figure 1) for their antiproliferative, apoptosis-inducing, and resistance-breaking properties.

We and others have recently reported on ruthenium(+2/+3) complexes with N-heterocyclic carbene (NHC) ligands [19,20,21,22,23,24,25] and their potential as novel anticancer drugs. For complexes of the type [(p-cymene)(NHC)RuCl2], we observed strong cytotoxic effects when the NHC ligand contained benzyl side chains on the nitrogen atoms of the NHC ligand that facilitated an efficient cellular uptake [23]. Thus, complex HB320 triggered strong cytotoxicity in MCF-7 breast cancer and HT-29 colon carcinoma cells. The complex also inhibited the activity of or bound to thiol- and selenol-containing biomolecules. Interestingly, HB320 triggered toxicity in zebrafish embryos at concentrations higher than the half maximal inhibitory concentration (IC_50_) values for anticancer cell toxicity, indicating a certain degree of selectivity. Complex HB324 was recently described with a series of antibacterial ruthenium complexes, where it triggered moderate activity against some gram-positive pathogenic bacteria [19].

## 2. Results

### 2.1. Anti-Proliferative and Apoptotic Activity

The initial experiments on human B cell precursor leukemia (Nalm-6) cells (Table 1) showed the best cytotoxic activity for the substance HB324, a ruthenium complex with an additional methyl group as compared to HB320. This complex induced more than 50% apoptosis on B-cell precursor leukemia cells (Nalm-6) at a concentration of 5 µM (see Appendix A), while the second complex, HB320, induces around 10% apoptosis at 5 µM. The two ligands also showed activity, but only starting at a concentration of 10 µM (see Appendix A).

Therefore, further investigations were focused on the compound HB324. The ability of the substance to induce apoptosis in Nalm-6 cells, as mentioned above, was tested by modified cell cycle analysis with FACS Calibur flow cytometry after 72 h of incubation with different concentrations. HB324 can induce DNA-fragmentation, which indicates that apoptosis occurs in leukemic Nalm-6 cells in correlation with the increasing concentrations. AC_50_, the concentration of the substance necessary to induce apoptosis in half of the cell population, was around 4 µM for HB324. HB324 triggered apoptosis induction reaches a plateau at concentrations above 10 µM (see Appendix A).

Besides apoptosis, there is a second main cell death mechanism, namely necrosis, that leads to an unintended non-specific cytotoxic effect [26]. A central task of an anticancer drug is the induction of apoptosis in malignant cells without triggering high rates of necrosis, which leads to the release of cell components into the extracellular space, can cause inflammatory processes, and prohibits the removal of cell debris by phagocytes [27].

To determine the mechanism triggered by the compound, an annexin-V/propidium iodide double staining and a LDH (lactate dehydrogenase) cytotoxicity assay were performed. As shown in Figure 2a, after 48 h of incubation, almost all cells showed late apoptosis, and within the tested biological active concentration range, no nonspecific necrotic effects could be detected up to a concentration of 20 µM (Figure 2b).

To test the in vitro anti-proliferative effect of HB324, Nalm-6 cells were incubated with different concentrations of the substance and solvent treated cells (0.5% dimethyl sulfoxide (DMSO)) served as the control. The overall inhibition of proliferation was determined by comparing the total amount of (vital) cells treated with the substances with the control. The results showed that the compound can effectively inhibit cell proliferation in leukemia Nalm-6 cells in a dose-dependent manner (Figure 2c). A concentration of 1 µM of HB324 caused 100% proliferation inhibition, which indicates an G1 arrest.

### 2.2. Inducing Apoptosis via the Intrinsic Pathway

There are two pathways by which apoptosis can occur, the intrinsic (also referred to as “mitochondrial pathway”) and extrinsic pathway, with different mechanisms of activation and connections between them [28]. Cells showing a reduced mitochondrial membrane potential give an indication that these cells go into apoptosis via the intrinsic pathway [29]. To test whether this is triggered by HB324, we performed JC-1 staining followed by flow cytometric measurement. Nalm-6 cells show a HB324-dose-dependent depolarization of the mitochondrial membrane potential (Figure 3a), thus, this compound triggers apoptosis via the intrinsic pathway. To further test the involvement of the extrinsic pathway, we incubated the compound for 72 h in BJAB mock (Burkitt-like lymphoma) cells and their sub-cell line BJAB FADD^−/−^, expressing a dominant negative FADD (FAS-associated death domain protein) mutant that is lacking the death domain [30]. The results show that the compound induces even more apoptosis on the modified cells than on the control cells (Figure 3b), which suggests that the induced apoptosis occurs independently of the CD95/Fas death receptor, meaning the extrinsic pathway. The dependence of the substance on components of the cell cycle or apoptosis, such as certain proteins, was tested by various experiments.

By Western blot analysis, the activation of caspase-3, -8, and -9 could be detected (Figure 3c). The results show a stable distribution of procaspase-8, i.e., no cleavage to the initiatorcaspase-8 of the extrinsic pathway [31], and thus providing further proof of the independence of this mechanism. Furthermore, a decrease, and thus cleavage, of procaspase-9 could be shown. This is an additional indication of the intrinsic apoptosis pathway, since procaspase-9 is part of the so-called apoptosome in the intrinsic pathway, and after its cleavage, effector caspases are activated and apoptosis is induced [32].

Harakiri (HRK), a product of the *harakiri* gene (*HRK*), is a pro-apoptotic Bcl-2 (B-cell lymphoma 2) homology domain 3-only protein of the Bcl-2 family [33] that interacts with and inhibits the anti-apoptotic and death-repressor proteins Bcl-2 and Bcl-xL (B-cell lymphoma-extra large) [34]. The activation of HRK by HB324 that is shown by Western blot analysis (Figure 3c) should be further verified by a cell model. The daunorubicin-resistant subcellline (NiWi-Dau) of chronic myeloid leukemia cells K562 has been shown in previous studies to under-express HRK [9]. The experiment showed that the complex cannot cause any effect on the cells (Figure 3d) with this mutation. This indicates that the effect caused by HB324 is dependent on the Harakiri protein or is inhibited by the Bcl-2 family proteins that lack the inhibition by Harakiri.

The protein Bcl-2 is the product of the anti-apoptotic gene *BCL2* that is mainly localized in the outer mitochondrial membrane and can prevent apoptosis through various interactions in the intrinsic pathway [35,36,37,38]. The vincristine-resistant sub-cell line of BJAB lymphoma cells (BiBo) showed overexpression of this anti-apoptotic protein in previous experiments. Comparison of apoptosis induction by HB324 on BJAB and BiBo cells reveals a significantly lower effect on the resistant cells (Figure 4a). In addition to further evidence for the intrinsic pathway, these results also suggest a dependence on Bcl-2 as HB324-induced apoptosis can be inhibited by Bcl-2 overexpression. Moreover, a clear cleavage of procaspase-3 into caspase-3 of apoptosis was detected in the Western blot analysis (Figure 3c). As an effector caspase it is part of the common end pathway of the two apoptosis pathways [32].

Incubation of HB324 on BJAB and doxorubicin-resistant BJAB cells (7CCA) expressing a reduced level of caspase 3 shows no apoptosis induction in the altered cells (Figure 4b). This indicates that apoptosis induction by HB324 is dependent on sufficient levels of caspase-3. A further indication of caspase dependence may be provided by an experiment with the pancaspase inhibitor Z-VAD-FMK. Incubation of Nalm-6 cells with 5 µM HB324 without and with pretreatment of the cells with the inhibitor, respectively, showed a significantly reduced apoptosis rate after caspase inhibition (Figure 4c).

To investigate apoptosis induction by HB324 via another mechanism, namely ROS (reactive oxygen species)-mediated, we used N-acetylcysteine (NAC) as a known ROS inhibitor and H_2_O_2_, which belongs to the ROS [39], as a positive control. Figure 4d shows that the ROS inhibitor NAC can significantly attenuate apoptosis induction by the ruthenium complex. Based on the reactivity of ruthenium complexes towards biological thiols, the contribution of a partial inactivation of HB324 upon interaction with NAC cannot be excluded.

### 2.3. Overcoming Drug Resistance

In the therapy of oncological diseases, the most frequent problem is the development of resistance to the conventionally used cytostatic drugs [1,40]. Therefore, the main focus of oncological research is on finding novel substances that are able to overcome such resistances. Thus, it is interesting that the substance HB324 can induce apoptosis in different cell lines that are resistant to various cytostatics.

First, we incubated the compound with Nalm-6 cells and the prednisolone-resistant cell line NaKu. In addition to the desired resistance against prednisolone, a glucocorticoid also used in tumor therapy [41], the cells also developed cross-resistances to cytostatic drugs, among others, to the antimetabolites cytarabine (pyrimidine analog) and cladribine (purine analog); the anthracyclines idarubicin, daunorubicin, doxorubicin, and epirubicin; the nitrogen-lost compounds/oxazaphosphorines cyclophosphamide and ifosphamide; or the platinum-containing cytostatics cisplatin, carboplatin, and oxaliplatin. These resistances are overcome by HB324 and there are equally high apoptosis rates in both cell lines, which are even higher in the NaKu cells (Figure 5a).

Additionally, we tested the complex on cytarabine-resistant MaKo cells (co-resistances: fludarabine, cladribine, clofarabine, cyclophosphamide, and vincristine) [9], methotrexate-resistant JeBa cells (co-resistances: mitoxantrone, idarubicin, daunorubicin, doxorubicin, and epirubicin) [42], and etoposide-resistant JeFri cells (co-resistances: cytarabine and methotrexate) [9], each in comparison to the baseline Nalm-6 cell line. In all experiments, HB324 was able to overcome the resistant cells, similarly to that shown for the NaKu cells (see Appendix A).

To test resistance overcoming in another cell type, we incubated the neuroblastoma cells SK-N-AS and its subline LiOn with the substance. In addition to its cisplatin resistance, this cell line also shows resistance to carboplatin, cytarabine, and daunorubicin. In previous experiments, reduced expression of caspase-8 was shown to be a possible mechanism of resistance [9]. Remarkably, the ruthenium complex HB324 was able to overcome the cisplatin resistance of the LiOn cells (Figure 5b). These results also fit the data from the Western blot shown above (Figure 3c) and the described independence of caspase-8 and the extrinsic pathway.

### 2.4. Selectivity towards Malignant Cells

Another favorable property of a potentially cytostatic substance is the ability to induce apoptosis in the diseased cells, but not to damage the healthy cells, thus causing as few side effects as possible. To compare the apoptotic effect of HB324 in malignant cells with the effect in healthy human cells, we performed a selectivity test. The cell lines Nalm-6 (leukemia), BJAB (lymphoma), and non-malignant leucocytes from a healthy test-person were incubated with increasing concentrations of HB324 up to 20 µM for 72 h. Both malignant cell lines, Nalm-6 and BJAB cells, showed significantly higher apoptosis rates and reached an AC_50_ at less than 10 µM. In contrast to the malignant cell lines, the healthy cells (leucocytes) showed an apoptosis rate less than 10% in all tested concentrations (Figure 6), which demonstrates a strong selectivity of the compound for malignant cells.

### 2.5. Synergistic Effects with Established Cytostatic Drugs

An effective strategy in cancer treatment is the combination of multiple cytostatic drugs in order to exploit possible synergistic effects, improve therapy selectivity, and lower the likelihood of toxic side effects and development of resistance [43,44]. To investigate whether HB324 exhibits synergistic effects with commonly used cytostatic agents, we incubated it on Nalm-6 cells with the universally used cytostatic drugs vincristine and daunorubicin individually at low concentrations and in combination. HB324 in combination with the anthracycline daunorubicin achieves a moderate but clearly visible synergistic effect between +80% and +130% (see Appendix A) but with the vinka alkaloid vincristine, in all combinations, a synergistic effect above + 250% consistently appears (Figure 7).

## 3. Discussion

In this study, we investigated the ability of the ruthenium complex HB324 to inhibit malignant cell growth, induce apoptosis in various cell lines, overcome resistance, and achieve synergistic effects with established cytostatic drugs. Based on the results of our experiments presented above, various effects and different cellular targets of HB324 can be assumed (Figure 8).

Remarkably, the second ruthenium complex HB320 and the two associated metal-free ligands HB20 and HB24 showed significantly lower effects. It seems that the metal plays an important role for the good efficacy of the substance, because the second ruthenium complex HB320 shows a smaller effect than HB324 but works better than the metal-free ligands HB20 and HB24. It is conceivable that the metal atoms contained in the ligands interact with the mitochondria and the contained metal atoms of the mitochondrial proteins and thus destabilize the mitochondrial membrane and cause mitochondrial outer membrane permeabilization (MOMP), resulting in the release of the mitochondrial apoptosis proteins. The influences on further cellular processes, such as DNA intercalation, etc., is also proposed. Why HB324 acts better than HB320, which differs by only one methyl group, could be steric effects leading to different reaction and binding properties. However, these questions require further biochemical studies to identify the cellular targets and, if necessary, the preparation of further complexes chemically derivatized on different structures to investigate the structure-activity relationship in more detail.

For a long time, various ruthenium-containing complexes have been part of research into novel cancer therapies, especially as an alternative to the platinum-containing substances such as cisplatin, which is one of few metal-based cytostatic agents used in clinical cancer therapy to date [45]. So far, only three of these ruthenium complexes have reached the stage of clinical trials, with only the two ruthenium(III) complexes being developed as chemotherapeutic drugs [18,46,47]. Consequently, the approach of using ruthenium complexes in the treatment of cancer is already showing success in clinical trials, however, there remains a need to develop, research, and further investigate novel promising ruthenium complexes, such as our ruthenium(II) complex, to achieve similar or better effects than the established drugs cisplatin, carboplatin, or oxaliplatin, which are most likely due to their metal content, but to improve or prevent the therapy-limiting toxic side effects when using these new substances.

A typical characteristic of tumor cells is rapid proliferation, which is why an important property of a potential anticancer drug is the growth inhibition of these cells. HB324 showed clear proliferation inhibition of leukemia cells in the low micromolar range. An induced G1 arrest and even a decrease of the cell number after 24 h already gave an indication that the substance can induce cell death in these cells.

Apoptosis is a programmed cell death that leads, among other events, to a translocation of phosphatidylserine to the outside of the cell membrane and a fragmentation of the DNA [28]. These two phenomena were triggered by HB324 in various cell types, such as acute and chronic leukemia cells, lymphoma cells, or neuroblastoma cells, and are thus proof of substance-induced apoptosis as the preferred cell death mechanism. These HB324-induced effects appear to be at least partially caspase-dependent, mainly on caspase-3, and seems to be triggered by several pathways. First, we identified the main apoptotic pathway used by the complex. Both the measurement of mitochondrial membrane potential and the dependence of the anti-apoptotic mitochondrial protein Bcl-2 indicates the use of the intrinsic apoptotic pathway. In addition, the independence of proteins of the extrinsic pathway, such as CD95/Fas death receptor or caspase-8, also indirectly suggests this. However, the generation and action of ROS could also be shown to be a part of the apoptosis mechanism of the ruthenium complex. Especially notable was the activation of the pro-apoptotic protein Harakiri by the compound and the significant dependence of apoptosis induction by the complex on the resistant chronic leukemia cells lacking this protein. This effect can be explained either by a direct dependence of apoptosis induction on the Harakiri protein, or by the lack of inhibition of the anti-apoptotic Bcl-2 family proteins due to absence of this protein. Either way, the apoptosis-promoting protein HRK appears to play a central and important role in the mechanism of action of the ruthenium complex.

Often, the use of metal-containing substances in cancer therapy such as cisplatin or carboplatin and oxaliplatin is limited by the cytotoxic side effects and disadvantageous effects on healthy cells like nephrotoxicity, ototoxicity, or gastrointestinal symptoms [15]. One such unintended non-specific cytotoxic effect is necrosis, which can lead to a (local) inflammatory reaction due to the swelling, subsequent bursting of the cells, and the leakage of cell contents such as cell organelles into the extracellular space [26,27]. This effect on the cells following the application of HB324 could be excluded up to a high micromolar range. In addition, the ruthenium complex showed beneficial selectivity against malignant cells such as leukemia and lymphoma cells without damaging healthy leukocytes, which may additionally reduce possible expected side effects. Many conventional cytostatic drugs exploit the faster proliferation of cancer cells compared to normal cells. However, since they do not limit the toxic effects to the malignant cells alone, therapy is often limited by the lack of selectivity and the associated side effects [48]. One reason why HB324 shows such outstanding selectivity for malignant cells may be that, as shown in our experiments, the complex addresses multiple targets within cells, such as ROS induction or upregulation of Hrk. ROS, such as O_2_-or H_2_O_2_, play an important role in various cellular functions such as proliferation and are physiologically generated in the mitochondrial respiratory chain [48,49,50]. In order to maintain the balance and not to exceed the physiological amount of ROS, the healthy cell possesses antioxidants, such as the superoxide dismutase (SOD) [48,50,51]. If the oxidative stress and the amount of ROS in a cell increases, this can lead to DNA damage, excessive proliferation, and genetic instability or mutations [48,50,52,53,54,55,56]. As has been shown in various studies, most cancer cells appear to produce elevated levels of ROS, which seems to account for the cancer cell phenotype. This leads to increased cell growth and proliferation, oncogenetic transformations, and can also lead to resistance to anticancer drugs. However, excessive levels of ROS lead to damage to various cellular components, damage to the mitochondrial membrane, and apoptosis by overloading the cellular antioxidant capacity of the cells [48,49,50,51,53,56]. Therefore, a widely accepted theory is that ROS increasing substances, such as presumably HB324, then act selectively on malignant cells, since the already initially increased ROS levels make the malignant cells more susceptible to a ROS increase and their antioxidative capacity is exhausted more quickly. It would also be conceivable that this mechanism could lead to sensitization to the other cellular targets of the compound. For example, ROS-enhanced DNA damage could lead to facilitated and increased induction of the apoptosis cascade, or ROS-destabilized mitochondrial membranes could lead to an increased susceptibility for the effect of increased Harakiri levels.

Another strategy in modern chemotherapy to reduce side effects and also prevent the development of resistance is the combination of two or more substances with different modes of action as polychemotherapy [15,57]. Through this, lower concentrations of the individual drugs are required and the selection of single resistant cells to one mechanism of action is prevented, while possible synergistic effects are exploited. Our investigated substance shows notable synergistic effects with established cytostatic drugs in low micromolar concentrations. The synergistic effects with conventional cytostatic drugs could possibly be due to the different targets of HB324 and the drugs. It is conceivable that the low concentrations lead to a respective sensitization of the cells to the effect of the other substances. Both vincristine and daunorubicin lead to inhibition of proliferation, mitosis, and replication, through inhibition of microtubules and intercalation with DNA, whereas HB324 appears to primarily affect the mitochondria directly.

One of the biggest problems in cancer treatment is the development of resistance to the cytostatic drugs used in chemotherapy [40,58,59,60,61]. Therefore, there is a particular need to develop and investigate new substances that can overcome these resistances and still induce apoptosis in these altered cells. The ruthenium complex HB324 can overcome various resistances against many different cytostatics, especially in leukemia cells. The development and investigation of substances that can overcome these resistances is one of the most important and main tasks in oncological research in the field of metal-based substances. Thus, overcoming cisplatin resistance in the neuroblastoma cell line LiOn deserves special mention, as the use of cisplatin remains the main option in some clinical situations, but its use often leads to resistance to cisplatin and consequently to treatment failure [62]. Without further biochemical studies, we cannot yet say exactly why HB324 is able to overcome these resistances. This would require more precise identification of the cellular targets of the substance and the binding properties to them, as well as monitoring of the cellular mechanisms triggered by the substance. By producing further complexes or modified derivatives, individual effects could be improved and, after more precise identification of the effect of individual chemical structures in the cells, the properties of the complexes could be adapted and improved. This could help to prevent metal-associated and unintended side effects as already known from the use of cisplatin or other metal-based substances, that can occur in potentially following animal experiments or clinical studies.

## 4. Materials and Methods

### 4.1. Chemicals and Drugs

All chemicals and solutions used for synthesis were purchased from common suppliers. Propidium iodide (PI; 50 µg/mL) and dimethyl sulfoxide (DMSO) were delivered from Serva (Heidelberg, Germany), RNase A from Qiagen (Hilden, Germany), ethanol absolute from Sigma-Aldrich (St.Louis, MO, USA), Formaldehyde solution 37 % from Carl Roth GmbH (Karlsruhe, Germany) and PBS (phosphate-buffered saline) from GIBCO invitrogen (Karlsruhe, Germany).

The primary IgG antibodies anticaspase-3, -8, and -9 were purchased from Santa Cruz Biotechnology (Dallas, TX, USA), anti-harakiri from GeneTex (Irvine, CA, USA), and anti-ß-actin from Sigma-Aldrich (St.Louis, MO, USA). The secondary IgG antibodies, anti-mouse and anti-rabbit, were received from Bioscience (San Diego, CA, USA) and Promega (Minneapolis, MN, USA), respectively.

The broad-spectrum caspase inhibitor I (Z-VAD-FMK) was from Calbiochem (Merck, Darmstadt, Germany). NAC (N-acetyl-L-cysteine), used as a ROS (reactive oxygen species) inhibitor, was acquired from Sigma-Aldrich (St.Louis, MO, USA).

The conventional cytostatic drugs (daunorubicin, vincristine, doxorubicin, prednisolone, etoposide, methotrexate, cisplatin, cytarabine) were provided by Helios Clinics Schwerin and the children’s hospital Amsterdamerstraße (Cologne, Germany); they were dissolved in DMSO as stock solution directly before use in the experiments.

HB20, HB24, HB320, and HB324 were prepared as previously reported [19,23]. The investigated substances were dissolved in DMSO to give a stock solution of 40 mM.

### 4.2. Cell Lines and Cell Culture

The Nalm-6 cell line (human B cell precursor leukemia, ALL (acute lymphoblastic leukemia)) was kindly provided by Dr. Seeger/AG Henze, Charité Berlin, Germany. Cytarabine-(MaKo), prednisolone-(NaKu), etoposide-(JeFri), and methotrexate-resistant (JeBa) Nalm-6 cells were generated by our group by exposure to increasing concentrations of the respective cytostatic drugs. Compared to the origin cell line, the resistant cells tolerate significant concentrations of the cytostatic drugs without the loss of vitality.

The BJAB mock (Burkitt-like lymphoma) and BJAB FADDdn (FADD^−/−^) cells were kindly provided by Prof. Dr. P.T. Daniel, Charité Berlin. FADDdn cells are transfected with pcDNA3-FADD^−/−^. Consequently, they are expressing a dominant negative FADD (FAS-associated death domain protein) mutant that is lacking the death domain. The BJAB mock cells contain a pcDNA3-Primer without the FADDdn-Gen. Similar to the Nalm-6 cells, BJAB lymphoma cells were conditioned by treatment with vincristine (BiBo) and doxorubicin (7CCA) in increasing doses while maintaining a high level of vitality to generate the resistant cell lines.

Furthermore, SK-N-AS neuroblastoma cells were kindly provided by Professor Dr. T. Simon, University Cologne, Germany, and the cisplatin-resistant subline (LiOn) was generated by our group.

Moreover, immortalized K562 (chronic myelogenous leukemia (CML)) cells were kindly donated by PD Dr. Dr. Seeger Charité Berlin, Germany, and a daunorubicin-resistant subline (NiWi-Dau) was also generated in our group.

Leucocytes from a healthy test person were attained from a blood sample.

Cell cultures were maintained in an incubator in atmosphere of 5% CO_2_ and air at 37 °C. Twice times a week the cells were passaged, their vitality was controlled, and they were diluted to a concentration of 1 × 10^5^ cells/mL. All suspension cells and the adherent neuroblastoma cells were cultured in completed RPMI 1640 medium (GIBCO invitrogen, Karlsruhe, Germany), supplemented with heat-inactivated fetal bovine serum (FBS, 10% *v*/*v*, GIBCO invitrogen, Karlsruhe, Germany), 100 U/mL penicillin, and 100 µg/mL streptomycin (GIBCO invitrogen, Karlsruhe, Germany). For the leucocytes, RPMI 1640 medium was supplemented with 20% FBS (*v*/*v*, GIBCO invitrogen, Karlsruhe, Germany). Twenty-four-hours in advance of all experiments, the cell lines were adjusted to a concentration of 3 × 10^5^ cells per mL to guarantee controlled growth conditions. Right before the treatment with the respective substance, the cells were diluted to 1 × 10^5^ cells per mL.

### 4.3. Isolation of Healthy Human Leucocytes

Ten-milliliters RPMI 1640 medium (20%) was added to 50 mL of fresh blood taken from a healthy test person donated by the authors of this paper. Leukocytes were separated from the other blood components using Ficoll (Bicoll separating solution, Biochrom, Merck, Darmstadt, Germany) and density gradient centrifugation (18 min, 2000 rpm, 18 °C). The leukocytes were collected from the buffy coat and RPMI 1640 medium (20%) was added before centrifugation at 1500 rpm and 18 °C for 5 min. The cell count and viability were determined by the CASY Cell Counter and Analyzer System (OMNI Life Science GmbH, Bremen, Germany). Cells were seeded at a density of 3 × 10^5^ cells per mL prior to the experiment.

### 4.4. Determination of Cell Concentration and Viability

Cell concentration and viability was determined by using CASY Cell Counter and Analyzer System from OMNI Life Science GmbH (Bremen, Germany). Settings were specifically defined for the requirements of each used cell line. In one measurement, three size ranges are determined. The cell concentration is separated in cell debris, dead cells, and viable cells.

To determine the inhibitory effects of the analyzed substance, the cells were seeded at a concentration of 1 × 10^5^ cells per mL and treated with different concentrations of the substance; non-treated and solvent treated (0,5% DMSO) cells served as the controls. After 24 h of incubation at 37 °C and 5% CO_2_, cells were resuspended and 100 µL of each well was diluted in 10 mL CASYton (ready-to-use isotonic saline solution, OMNI Life Science GmbH, Bremen, Germany) for immediate automated counting. The increase in the cell number of exposed cells is determined and compared to the non-exposed control cells.

### 4.5. LDH Release Assay

The lysis of cells caused by a necrotic cell death can be quantified by the detection of lactate-dehydrogenase (LDH) release from the cytosol to the medium [63]. The LDH activity is determined by a linked enzymatic test. In a first step, NAD+ is reduced to NADH/H+ during the conversion of lactate to pyruvate catalyzed by LDH. Then, a catalyst transfers a H/H+ equivalent to the pale-yellow tetrazolium salt forming a red formazan salt, which is detected at 490 nm.

The release of lactate-dehydrogenase is directly connected with the rupture of cell membrane caused by a necrotic cell death. In contrast to the relatively slow process of apoptosis the effect of necrosis occurs directly after the treatment of the cells with the substance. Therefore, necrosis and apoptosis can be differentiated by the rapid detection of LDH activity in the extracellular medium. The lactate-dehydrogenase is quantified in cell culture supernatants after over 1 h of incubation of the cells with the substance using the Cytotoxicity Detection Kit from Roche (Basel, Switzerland).

The cells were centrifuged at 1500 rpm for 10 min. Cell-free supernatants (100 µL) and the reaction mixture (100 µL) was mixed and incubated for 20 min at room temperature. Then, the time-dependent formation of the reaction product was quantified photometrically at 490 nm. The maximum amount of LDH activity released by the cells was determined by lysis with Triton X-100 (0.1%, Sigma-Aldrich, St.Louis, MO, USA) in culture medium and was set as 100% cell death.

### 4.6. Annexin-V/Propidium Iodide Double Staining

Phosphatidylserine (PS), which is normally located in the inner membrane of the cell, is translocated to the outside in the early phase of apoptosis and can be bound by annexin-V in a calcium-dependent manner [64]. Propidium iodide (PI) intercalates into the DNA and can only bind it when the membrane is permeable, i.e., in the late apoptotic or necrotic state. Thus, living cells are annexin-V and PI negative, early apoptotic cells are annexin-V positive and PI negative, late apoptotic cells are annexin-V and PI positive, and necrotic cells are annexin-V negative and PI positive. To determine the number of apoptotic cells in the sample, the cells were stained with annexin-V conjugated with the fluorescent dye fluorescein isothiocyanine (FITC) and PI from the Annexin-V-FLUOS Staining Kit (Roche, Basel, Switzerland), and analyzed by flow cytometry.

Cells were seeded at a density of 1 × 10^5^ cells per mL and treated with different concentrations of the compound. After incubation for 48 h at 37 °C and 5% CO_2_, cells were collected by centrifugation (5 min, 4 °C, 8000 rpm) and washed in PBS. After re-pelleting, annexin-V-FITC/PI solution in incubation buffer was added to each sample. As controls, four samples were incubated with incubation buffer, annexin-V-FITC in incubation buffer, PI in incubation buffer or annexin-V-FITC and PI in incubation buffer for 10 min. This was followed by flow cytometric analysis by using a FACScan (Becton Dickinson, Heidelberg, Germany) equipped with the CELL Quest software.

### 4.7. Measurement of Mitochondrial Membrane Permeabilization

The mitochondrial permeability transition was determined by staining the cells with 5,5′, 6,6′-tetrachloro-1,1′,3,3′-tetraethylbenzimidazolylcarbo-cyanine iodide (JC-1; Molecular Probes, Leiden, the Netherlands). After incubation of the cells for 48 h at 37 °C with different concentrations of the substance, with an equal volume of DMSO (≤0.5%) or untreated cells as the control, the cells were collected by centrifugation at 3000 rpm, 4 °C for 5 min. Then, the cells were resuspended in 500 µL phenol red-free RPMI 1640 medium without supplements, and JC-1 was added to give a final concentration of 2.5 µg/mL (except the negative control). The cells were incubated for 30 min at 37 °C with moderate shaking. The cells were pelleted by centrifugation at 4000 rpm, 4 °C for 5 min, washed with ice-cold PBS, and resuspended in 200 µL PBS at 4 °C. Mitochondrial permeability transition was then quantified by flow cytometric determination of cells with decreased fluorescence-that is, mitochondria displaying a lower membrane potential (ΔΨm). Data were collected and analyzed using a FACScan (Becton Dickinson) equipped with the CELLQuest software. Data are given in percentage cells with low ΔΨm, which reflects the number of cells undergoing mitochondria-dependent apoptosis.

### 4.8. DNA Fragmentation

Apoptotic cell death was determined by a modified cell cycle analysis, which detects DNA fragmentation on the single cell level as described previously [65]. Cells were seeded at a concentration of 1 × 10^5^ cells per mL and treated with different concentrations of the substance, equivalent volumes of DMSO, or left untreated as the control in three replicate wells. After 72 h of incubation at 5% CO_2_ and 37 °C, cells were collected by centrifugation at 1500 rpm for 5 min and washed with PBS at 4 °C. After fixation in PBS/2% (*v*/*v*) formaldehyde on ice for 30 min, cells were pelleted, incubated with ethanol/PBS (2:1, *v*/*v*) for 15 min, pelleted, and resuspended in PBS containing 40 µg/mL RNase A. RNA was digested for 30 min at 37 °C and then cells were pelleted again. Finally, the cells were resuspended in PBS containing 50 µg/mL propidium iodide. Nuclear DNA fragmentation was quantified by flow cytometric determination of hypodiploid DNA. Data were collected and analyzed using a flow cytometry analysis by FACScan (Becton-Dickinson, Heidelberg, Germany) equipped with CELL Quest software. Data are given in percent of hypodiploidy (subG1), which reflects the number of apoptotic cells. The induced apoptosis is determined by subtracting the background apoptosis measured in the control cells from the measured apoptosis of the treated cells.

### 4.9. Western Blot Immunoblotting

After incubation for 36 h with different concentrations of the substance, daunorubicin at 76 nM as the positive control, the equivalent volume of DMSO, and untreated cells as the control, Nalm-6 cells were washed twice with PBS and lysed in buffer containing 10 mmol/L Tris-Hcl, pH 7.5, 300 mmol/L NaCl, 1% Triton X-100, 2 mmol/L MgCl_2_, 5 µmol/L EDTA, 1 µmol/L pepstatin, 1 µmol/L leupeptin and 0.1 mmol/L phenylmethylsulfonyl fluoride (PMSF). Protein concentration was determined using the bicinchoninic acid assay (Smith, Krohn et al. 1985) from Pierce (Rockford, IL, USA), and equal amounts of protein (20 mg per lane) were separated by sodium dodecyl sulfate–polyacrylamide gel electrophoresis (SDS-PAGE; Laemmli 1970). Immunoblotting was performed as described. Membranes were swollen in CAPS-buffer (10 mmol/L 3-[cyclohexylamino]propane-1-sulfonic acid, pH 11, 10% MeOH) for several minutes, and blotting was performed at 1 mA/cm^2^ for 1 h in a transbolt SD cell (BioRad, München, Germany). The membrane was blocked for 1 h in PBST (PBS, 0.05% *v*/*v* Tween-20) containing BSA (bovine serum albumin, Sigma-Aldrich, St.Louis, MO, USA) and incubated with a primary antibody for 1 h. The primary IgG antibodies anticaspase-3, -8, and -9 were diluted 1:1000 in PBST. Afterwards, the membrane was washed in PBST three times, and a secondary antibody was applied for 1 h. The secondary IgG antibodies anti-mouse and anti-rabbit were diluted 1:2000 and 1:2500 in PBST, respectively. After washing the membrane with PBST again, the protein bands were detected using ECL enhanced chemiluminescence system (Amersham, Braunschweig, Germany).

### 4.10. Statistics and illustration

The results shown correspond to a triplicate determination. The standard deviations (SDs) are shown as error bars. If SD is very small, error bars may not be visible. Graphs were drawn and statistics calculated by using Microsoft Office Excel. Parts of Figure 8 were drawn by using pictures from Servier Medical Art. Servier Medical Art by Servier is licensed under a Creative Commons Attribution 3.0 Unported License (https://creativecommons.org/licenses/by/3.0/). The other parts of the illustration were created with Microsoft Office PowerPoint.

## 5. Conclusions

Taken together, the results of our investigations of the ruthenium(II) complex HB324 show promising properties for a potential agent in cancer therapy, also with regard to the need of metal-based substances as an alternative to established platin-based chemotherapy. In contrast to the other complexes and their metal-free ligands, this complex showed good effects even in low micromolar concentrations, especially regarding proliferation inhibition and apoptosis induction. Furthermore, mainly mechanistic studies should follow to identify further properties of the complex and other effects in various cell lines and more combinations, e.g., apoptosis and cell cycle specific genes and proteins.

## Figures and Tables

**Figure 1 ijms-24-00952-f001:**
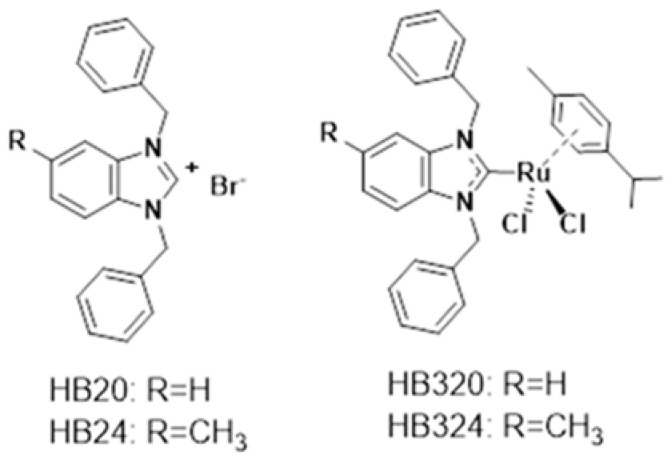
Ruthenium complexes and their benzimidazolim precursors investigated in this report.

**Figure 2 ijms-24-00952-f002:**
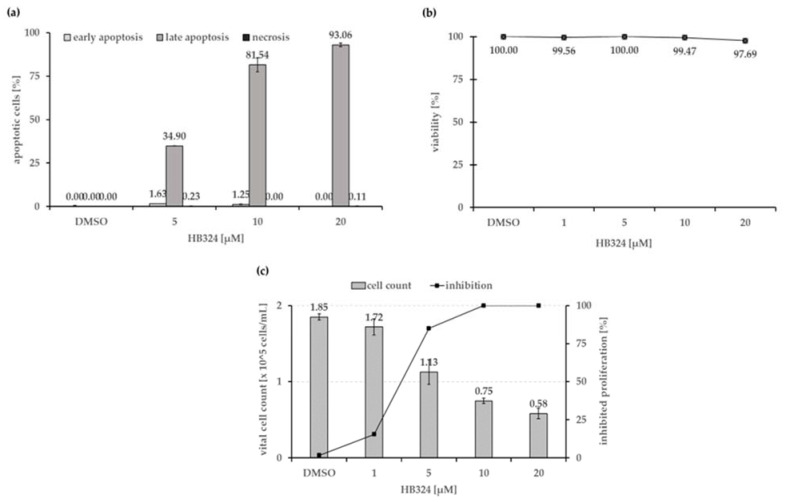
(**a**) Nalm-6 cells were treated and incubated with increasing concentrations of HB324 for 48 h. Solvent treated cells (0.5% dimethyl sulfoxide (DMSO)) served as the control. Distribution of counted cells according to cell status was measured by flow cytometric analysis. Vital cells, representing the remaining fraction, are not shown. Depicted values are mean values ± SD (n = 3). (**b**) To exclude unspecific, cytotoxic effects as necrotic cell death, the viability of Nalm-6 cells was determined by measurement of LDH release into the medium after 1 h of incubation with different concentrations of the compound. A proportion of cells were treated with 0.5% DMSO as the solvent control. Values are given as % of control ± SD (n = 3). (**c**) Nalm-6 cells were seeded at a density of 1 × 10^5^ cells/mL, and proliferation was measured after 24 h of incubation with the complex using the CASY^®^ CellCounter and Analyzer System. Solvent treated cells (0.5% DMSO) served as the control. Inhibition of proliferation is given in % of control as mean values ± SD (n = 3).

**Figure 3 ijms-24-00952-f003:**
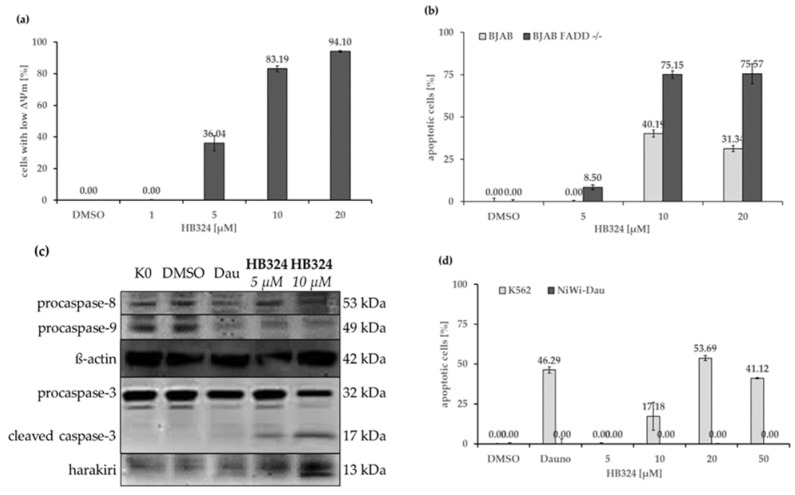
(**a**) Reduction of mitochondrial membrane potential measured by flow cytometric analysis of Nalm-6 cells after 48 h incubation with different concentration of HB324 and staining the cells with the cationic dye JC-1. Solvent treated cells (0.5% DMSO) served as the control. Values of the mitochondrial permeability transition are given as percentages of cells with low ∆Ψm as mean values ± SD (n = 3). (**b**) BJAB and BJAB FADD^−/−^ cells were treated for 72 h with three concentrations of the compound. Solvent treated cells (0.5% DMSO) served as the control. Nuclear DNA fragmentation was quantified by flow cytometric determination of hypodiploid DNA. Data are given in % hypoploidy (subG1), which reflects the number of apoptotic cells as mean values ± SD (n = 3). (**c**) Nalm-6 cells were incubated for 36 h with HB324 (5 µM and 10 µM) and 76 nM daunorbicin (Dau) as the positive control. Untreated and solvent treated cells (0.5% DMSO) served as the control. Proteins were separated by sodium dodecyl sulfate–polyacrylamide gel electrophoresis (SDS-PAGE) and subjected to Western blot analysis. Immunoblots developed with anticaspase-3, -8, -9 and antiharakiri is shown. Equal loading and blotting were verified by detection of ß-actin (42 kDa). (**d**) K562 and NiWi-Dau cells were treated for 72 h with different concentrations of the compound. Solvent treated cells (0.5% DMSO) served as the control and 2 µM daunorubicin (Dauno) was used as the positive control to prove the resistance. Nuclear DNA fragmentation was quantified by flow cytometric determination of hypodiploid DNA. Data are given in % hypoploidy (subG1), which reflects the number of apoptotic cells as mean values ± SD (n = 3).

**Figure 4 ijms-24-00952-f004:**
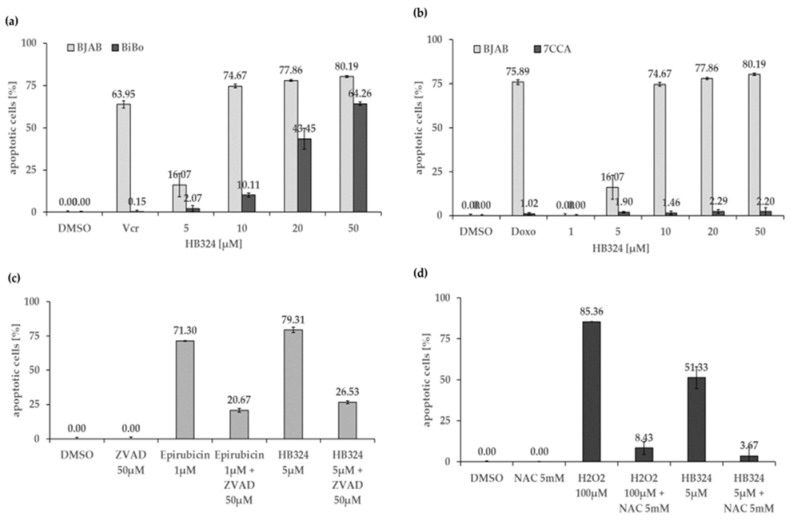
(**a**) Apoptosis induction by HB324 in BJAB and vincristine-resistant BJAB (BiBo) cells after 72 h. Solvent treated cells (0.5% DMSO) served as the control and 5.6 nM vincristine (Vcr) served as the positive control. Values of DNA fragmentation are given as mean percentages of cells with hypodiploid DNA ± S.D (n = 3). (**b**) BJAB and doxorubicin-resistant BJAB (7CCA) cells were treated with HB324 for 72 h. Solvent treated cells (0.5% DMSO) served as the control, and 184 nM doxorubicin (Doxo) was pipetted in both cell lines to prove the resistance. Values are given in percentage of apoptotic cells as mean values ± SD (n = 3). (**c**) Nalm-6 cells were incubated for 72 h with 1 µM epirubicin as the positive control or 5 µM HB324 after pretreating part of the cells with 50 µM of the pancaspase inhibitor Z-VAD-FMK (ZVAD) 1 h prior. Solvent treated cells (0.5% DMSO) served as the control. Nuclear DNA fragmentation was quantified by flow cytometric determination of hypodiploid DNA. Data are given in % hypoploidy (subG1), which reflects the number of apoptotic cells as mean values ± SD (n = 3). (**d**) Nalm-6 cells were incubated for 72 h with 100 µM H_2_O_2_ as the positive control or 5 µM HB324 after pretreating part of the cells with 5 mM of N-acetylcysteine (NAC) 1 h prior. Solvent treated cells (0.5% DMSO) served as the control. Nuclear DNA fragmentation was quantified by flow cytometric determination of hypodiploid DNA. Data are given in % hypoploidy (subG1), which reflects the number of apoptotic cells as mean values ± SD (n = 3).

**Figure 5 ijms-24-00952-f005:**
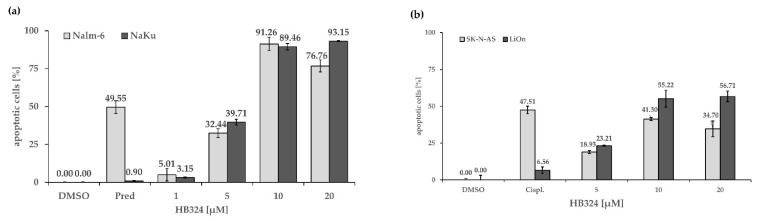
(**a**) Apoptosis induction by HB324 in Nalm-6 and prednisolone-resistant Nalm-6 cells (NaKu). After 72 h of incubation with different concentrations of the agent DNA fragmentation was measured via FACS scan analysis. Solvent treated cells (0.5% DMSO) served as the control, and 55.6 µM prednisolone (Pred) served as the positive control. Values of DNA fragmentation are given as mean percentages of cells with hypodiploid DNA ± SD (n = 3). (**b**) SK-N-AS and LiOn cells were treated with 5 µM, 10 µM, and 20 µM of HB324. Solvent treated cells (0.5% DMSO) served as the control, and 7.5 µM cisplatin (Cispl.) was pipetted in both cell lines to prove the resistance. After 72 h of incubation, DNA fragmentation was measured by flow cytometric analysis. Values are given as % of cells with hypodiploid DNA as mean values ± SD (n = 3).

**Figure 6 ijms-24-00952-f006:**
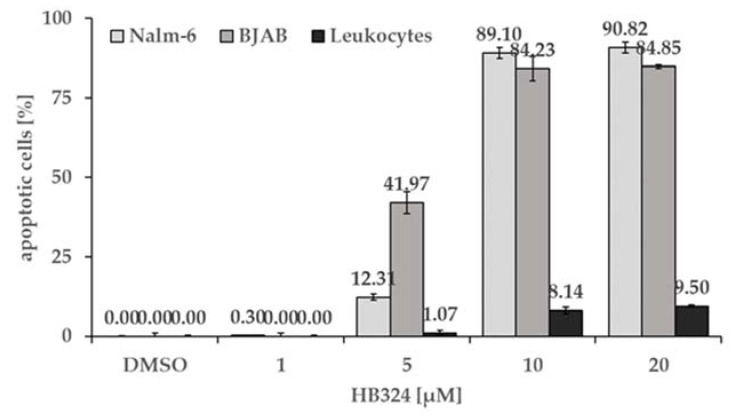
Nalm-6 cells, BJAB cells, and human leukocytes were treated with different concentrations of the substance. Solvent treated cells (0.5% DMSO) served as the control. All cells were incubated for 72 h. Then, induction of apoptosis was measured by flow cytometric analysis of cellular content. Values are given as % of cells with hypodiploid DNA as mean values ± SD (n = 3).

**Figure 7 ijms-24-00952-f007:**
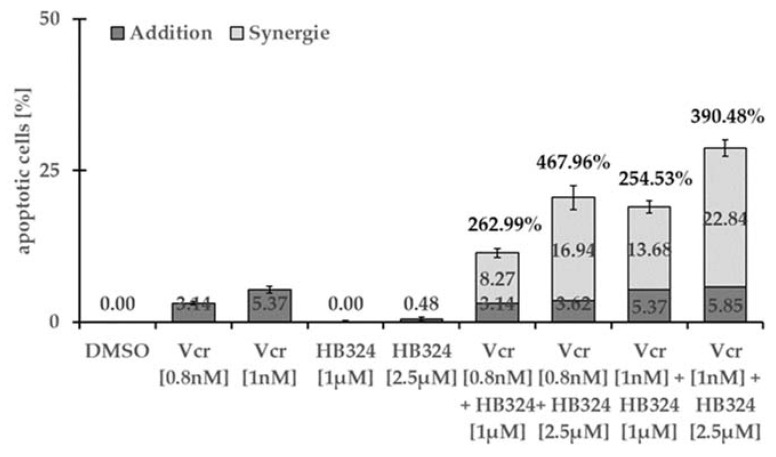
Nalm-6 cells were treated with 1 µM and 2.5 µM of HB324 and 0.8 nM and 1 nM of vincristine (Vcr) alone and in combination. Solvent treated cells (0.5% DMSO) served as the control. After 72 h of incubation, DNA fragmentation was measured by flow cytometric analysis. Values are given in percentage of apoptotic cells and as mean values ± SD (n = 3). The synergistic effect is shown as the percentages of the additional number of apoptotic cells relative to the sum of the individual number of apoptotic cells.

**Figure 8 ijms-24-00952-f008:**
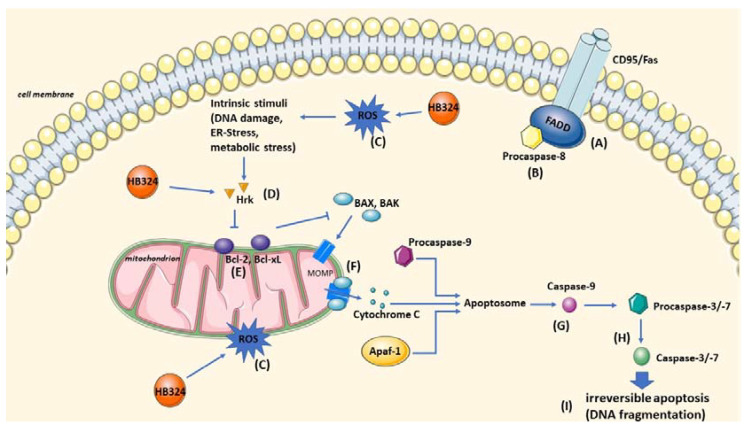
Illustration of the cellular targets and effects of HB324. Marked are the cellular targets identified by different detection methods in this paper. (**A**) FADD by cell pair model BJAB and BJAB FADD^−/−^. (**B**) Caspase-8 by Western blot and cell pair model SK-N-AS and LiOn. (**C**) ROS by apoptosis detection with and without the ROS inhibitor NAC. (**D**) Harakiri by Western blot and cell pair model K562 and NiWi-Dau. (**E**) Bcl-2 by cell pair model BJAB and BiBo. (**F**) Mitochondrial outer membrane permeabilization by JC-1. (**G**) Caspase-9 by Western blot. (**H**) Caspase-3 by Western blot and cell pair model BJAB and 7CCA. (**I**) Apoptosis by DNA fragmentation measurement. The Figure was partly generated using Servier Medical Art, provided by Servier, licensed under a Creative Commons Attribution 3.0 unported license.

**Table 1 ijms-24-00952-t001:** Compounds and ligands tested in this study for induction of apoptosis. Nalm-6 cells were incubated with each agent for 24 h and 72 h. Proliferation was measured after 24 h using the CASY^®^ CellCounter and Analyzer System and apoptosis was determined after 72 h using modified cell cycle analysis involving propidium iodide-staining. G1 arrest means a proliferation inhibition equal to 100% or above, indicating occurring cell death. AC_50_ is the concentration necessary to induce apoptosis in half of the cell population. Each value is the mean of three replicants ± standard deviation (SD).

Compound Name	G1 Arrest [µM]	AC_50_ [µM]
HB20	≥20 µM	>50 µM
HB24	≥10 µM	~20 µM
HB320	≥20 µM	~7.5 µM
HB324	≥1 µM	~4 µM

## Data Availability

The data presented in this study are available on request from the corresponding author.

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
