# Peer review of "Ruthenium Complex HB324 Induces Apoptosis via Mitochondrial Pathway with an Upregulation of Harakiri and Overcomes Cisplatin Resistance in Neuroblastoma Cells In Vitro"

_ijms, 2023, doi:10.3390/ijms24020952_

Round 1

Reviewer 1 Report

     Authors have demonstrated that the ruthenium complex HB324 promotes apoptosis via mitochondrial pathways in cancer cells and upregulates the Harakiri resistance protein that blocks the anti-apoptotic and death repressor proteins Bcl-2 and BCL-XL. Moreover, they have shown that this complex exerts a synergistic effect with several anticancer drugs and overcomes resistance in some cancer cell lines. Authors conclude that HB324 is a “promising potential as novel anticancer agent in-vitro, suggesting further investigations on this and other preclinical ruthenium drug candidates”.

     The manuscript is well written. The abstract summarizes the purpose of the study. The introduction covers the literature and explains the basis for this study. The methods are sound and accurately performed. The results are also a straight-forward description of the findings observed and they are supported with good figures. Discussion summarizes the present data and correlates these findings with appropriate information from the literature.

          However, some points must be improved:

1.      Discussion. This section is interesting but it would be improved if authors suggest hypothesis and/or explain in-depth as much as possible the findings reported. For example, see lines 335-337. Could authors suggest a hypothesis to explain why the ruthenium complex HB324 did not cause a damage in healthy leukocytes but a strong selectivity against malignant cells? What has to be done in the future regarding the research on the ruthenium complex HB324? Authors say: suggesting further investigations on this and other preclinical ruthenium drug candidates. This sentence must be developed: these investigations and the ruthenium drug candidates must be mentioned in the text and discussed. In the same way, see lines 324-325: “action of the ruthenium complex and should be verified and analyzed in more detail in further studies”. Mention and discuss these further studies. That is, authors must indicate the key research lines on ruthenium complexes that must be developed in the future. What drawbacks would there be in using the ruthenium complex in future clinical trials? Do the authors have any idea (e.g., chemical structure) why the best substance tested was HB324 and not HB20, HB24 or HB320? This is mentioned in lines 286-287 but not discussed. Any idea (e.g., chemical structure when compared with common used cytostatic drugs, upregulation of certain substances) about why does HB324 overcome resistance especially in leukemia cells? In sum, discuss in-depth as much as possible the findings reported and suggest future research lines.

2.      Figures. A figure showing how HB324 acts in cancer cells woul be welcome. For example, in this figure, the different signaling pathways involved in apoptosis (e.g., caspase 3, ROS) must appear. This figure will help to better explain how this compound promotes apoptosis, upregulates the Harakiri protein, overcomes resistance and exerts a synergistic effect.

3.      Minor point: decrease the number of paragraphs; join the text. For example, from line 28 to 42, one paragraph instead of 3 and so on along the text.

4.      References. Check list of references according to the Instructions for Authors.

     In sum, this is a competent study addressing a subject of interest and potential value for the scientific community, and it should be useful for interdisciplinary correlative studies.

Author Response

Response to Reviewer 1 comments

Comments and Suggestions for Authors

            Authors have demonstrated that the ruthenium complex HB324 promotes apoptosis via mitochondrial pathways in cancer cells and upregulates the Harakiri resistance protein that blocks the anti-apoptotic and death repressor proteins Bcl-2 and BCL-XL. Moreover, they have shown that this complex exerts a synergistic effect with several anticancer drugs and overcomes resistance in some cancer cell lines. Authors conclude that HB324 is a “promising potential as novel anticancer agent in-vitro, suggesting further investigations on this and other preclinical ruthenium drug candidates”.

            The manuscript is well written. The abstract summarizes the purpose of the study. The introduction covers the literature and explains the basis for this study. The methods are sound and accurately performed. The results are also a straight-forward description of the findings observed and they are supported with good figures. Discussion summarizes the present data and correlates these findings with appropriate information from the literature.

However, some points must be improved:

  1. Discussion. This section is interesting but it would be improved if authors suggest hypothesis and/or explain in-depth as much as possible the findings reported. For example, see lines 335-337. Could authors suggest a hypothesis to explain why the ruthenium complex HB324 did not cause a damage in healthy leukocytes but a strong selectivity against malignant cells? What has to be done in the future regarding the research on the ruthenium complex HB324? Authors say: suggesting further investigations on this and other preclinical ruthenium drug candidates. This sentence must be developed: these investigations and the ruthenium drug candidates must be mentioned in the text and discussed. In the same way, see lines 324-325: “action of the ruthenium complex and should be verified and analyzed in more detail in further studies”. Mention and discuss these further studies. That is, authors must indicate the key research lines on ruthenium complexes that must be developed in the future. What drawbacks would there be in using the ruthenium complex in future clinical trials? Do the authors have any idea (e.g., chemical structure) why the best substance tested was HB324 and not HB20, HB24 or HB320? This is mentioned in lines 286-287 but not discussed. Any idea (e.g., chemical structure when compared with common used cytostatic drugs, upregulation of certain substances) about why does HB324 overcome resistance especially in leukemia cells? In sum, discuss in-depth as much as possible the findings reported and suggest future research lines.

1. Response: Thank you for your comment. We have tried to present our assumptions about the effect of HB324 as clearly and comprehensibly as possible and to work out which further investigations are necessary for a better understanding of the effect of HB324 and for possible further developments and improvements.

  1. Figures. A figure showing how HB324 acts in cancer cells would be welcome. For example, in this figure, the different signaling pathways involved in apoptosis (e.g., caspase 3, ROS) must appear. This figure will help to better explain how this compound promotes apoptosis, upregulates the Harakiri protein, overcomes resistance and exerts a synergistic effect.

2. Response: We have tried to present the cellular points of attack and the presumed effects of HB324 as clearly and simply as possible (Figure 8) and have included this figure in the discussion.

  1. Minor point: decrease the number of paragraphs; join the text. For example, from line 28 to 42, one paragraph instead of 3 and so on along the text.

3. Response: The number of paragraphs has been reduced, leaving the paragraphs that mark a new section of text in terms of content and subject matter.

  1. References. Check list of references according to the Instructions for Authors.

4. Response: The reference list has now been completely revised and the suggested style file has been used.

      In sum, this is a competent study addressing a subject of interest and potential value for the scientific community, and it should be useful for interdisciplinary correlative studies.

Reviewer 2 Report

The manuscript entitled on"Ruthenium complex HB324 induces apoptosis via mitochondrial pathway with an upregulation of Harakiri and overcomes cisplatin resistance in neuroblastoma cells in vitro" by Nicola Wilke et.al tried to find the role of a potential drug in cancer treatment. The research is interesting. Before the manuscript is suggested for publication in the journal, some concerns need to be addressed.

1. Many abbreviations in the manuscript do not indicate their full name when they appear for the first time.

2. All the pictures in the manuscript are blurred, and the author should consider improving the pixels of the picture.

3. Figure 3 (c): the Western blots are of very poor quality.

4. Much of the background text on apoptosis, Harakiri(HRK) and bcl-2 in"2.2. Inducing apoptosis via the intrinsic pathway" are unnecessary and should be deleted.

5. The illustration of Figure 5(a) "Apoptosis induction induced by HB324 in Nalm-6 and prednisolon-resistant Nalm-6 cells (NaKu)." The words "induction" and "induced" are repeated.

6. There are 8 pictures (SI 1-SI 8) in the "supplement" data, but the corresponding pictures are not clearly marked in the results of the manuscript .

7. There are many wrong lines in the manuscript, such as 384 lines and 385 lines. Please check it carefully.

8. There are many inconsistencies in the format of "4.Materials and Methods". For example, the unit "mL" in "50 µg/" and "100 U/mL" should be unified. There are many irregularities in format, such as non-standard superscripts and subscripts, "CO2" should be changed to "CO2", "1x105" should be changed to "1x105", "1 mA/cm2" should be changed to "1 mA/cm2", and so on.

9. What primary antibody and secondary antibody? Specify.

Author Response

Response to Reviewer 2 comments

Comments and Suggestions for Authors

The manuscript entitled on "Ruthenium complex HB324 induces apoptosis via mitochondrial pathway with an upregulation of Harakiri and overcomes cisplatin resistance in neuroblastoma cells in vitro" by Nicola Wilke et.al tried to find the role of a potential drug in cancer treatment. The research is interesting. Before the manuscript is suggested for publication in the journal, some concerns need to be addressed.

  1. Many abbreviations in the manuscript do not indicate their full name when they appear for the first time.

1. Response: In our opinion, all necessary abbreviations have now been provided with the full name at the first mention.

  1. All the pictures in the manuscript are blurred, and the author should consider improving the pixels of the picture.

2. Response: All images in the enclosed zip folder now have the required size of minimum 100 pixels width/height.

  1. Figure 3 (c): the Western blots are of very poor quality.

3. Response: We apologize for the poor quality of the western blots. These are our original data and with our possibilities in the laboratory at present not differently or better to implement.

  1. Much of the background text on apoptosis, Harakiri (HRK) and bcl-2 in"2.2. Inducing apoptosis via the intrinsic pathway" are unnecessary and should be deleted.

4. Response: We have taken note of this comment. We believe that the background information contained therein is important for the understanding of the mechanism of action of HB324 and useful as a basis for the conclusions in the discussion.

  1. The illustration of Figure 5(a) "Apoptosis induction induced by HB324 in Nalm-6 and prednisolon-resistant Nalm-6 cells (NaKu)." The words "induction" and "induced" are repeated.

5. Response: Thank you for mentioning this, it has now been corrected.

  1. There are 8 pictures (SI 1-SI 8) in the "supplement" data, but the corresponding pictures are not clearly marked in the results of the manuscript.

6. Response: The pictures in the supporting information that are referred to in the main text have now been marked individually in the main text.

  1. There are many wrong lines in the manuscript, such as 384 lines and 385 lines. Please check it carefully.

7. Response: All wrong line breaks and paragraph marks we found have been corrected. However, there were not many of them, and further incorrect lines may be due to format changes and the use of different programs?

  1. There are many inconsistencies in the format of "4. Materials and Methods". For example, the unit "mL" in "50 µg/" and "100 U/mL" should be unified. There are many irregularities in format, such as non-standard superscripts and subscripts, "CO2" should be changed to "CO2", "1x105" should be changed to "1x105", "1 mA/cm2" should be changed to "1 mA/cm2", and so on.

8. Response: The different units of the quantities, for example of penicillin (100 U/mL) and streptomycin (100 µg/mL) are protocol-compliant specifications and have been used and published in our working group for years. All other units of quantity specifications are also manufacturer specifications or according to protocol.

The irregularities in the format of the subscripted and superscripted numbers in the method section have been corrected.

  1. What primary antibody and secondary antibody? Specify.

9. Response: The antibodies used have now been listed in section "4.9. Western blot immunoblotting", as well as the concentration of the dilutions prepared.

Information on suppliers can still be found in section "4.1 Chemicals and drugs".